# Where Is the Way Forward for New Media Empowering Public Health? Development Strategy Options Based on SWOT-AHP Model

**DOI:** 10.3390/ijerph191912813

**Published:** 2022-10-06

**Authors:** Zikang Hao, Mengmeng Zhang, Kerui Liu, Xiaodan Zhang, Haoran Jia, Ping Chen

**Affiliations:** 1Department of Physical Education, Laoshan Campus, Ocean University of China, 238 Song Ling Rd., Qingdao 266100, China; 2Department of Competitive Sports Center, Taishan Campus, Taishan College of Science and Technology, 223 Dai Zong Rd., Taian 271000, China; 3Department of Sports Medicine, Daiyue Campus, Shandong First Medical University, 619 Chang Cheng Rd., Taian 271000, China; 4Department of Journalism, Minsk Oblast, Belarusian State University, 4 Nezavisimosti Street, 220071 Minsk, Belarus

**Keywords:** public health, new media, social media, sports medicine, SWOT-AHP, sustainable development

## Abstract

(1) Background: In recent years, new media and the integration of sport and medicine have promoted the rapid integration and development of the two fields of health and, to a certain extent, the pursuit of public health knowledge and the promotion of health concepts. However, the overall development process is at an early stage and the aim of this paper is to make an empirical analysis of its development through a SWOT-AHP model and give corresponding recommendations. (2) Methods: The SWOT-AHP model was constructed to quantitatively and qualitatively analyse the four dimensions of strengths, weaknesses, opportunities and threats obtained through the Delphi method, with regard to development and to determine the strategic direction of its development. (3) Results: The strategic azimuth *θ* is −13.243° and the strategic coefficient *p* is 0.53699, in the diversification zone. (4) Conclusions: New media, as a fast track to empowering the integration of sport and medicine for health, is a field with a bright future, but its own strengths and external threats coexist and should be maximised to overcome the disruptions caused by external threats through a variety of measures.

## 1. Introduction

As society progresses, the material well-being of the public is being further enhanced, but the number of subhealth (the third state, also known as the grey state, pre-morbid state, sub-clinical stage, pre-clinical stage and latent disease stage) includes the absence of clinical symptoms or the perception of minimal symptoms, albeit with underlying pathological information. This concept is more popular in Chinese academia and society, and is considered by scholars to be similar to the “chronic fatigue syndrome” developed by the US Centers for Disease Control and Prevention [1]; chronically ill groups are gradually increasing due to the weakness of the concept of healthy living and the lack of knowledge about healthy living [2,3,4,5], even if the WHO does a good job on the social and commercial determinants of health [6,7].

The understanding that sport and medicine are integrated to promote health (a branch of medicine that promotes fitness and the prevention of injuries related to sport) is increasingly accepted and applied in a context where more and more researchers are confirming the considerable rehabilitative effects of sport for some diseases, and the range of applications is becoming broader and broader as people delve into the topic [8,9]. Sport and medicine are, to some extent, similar, in that they both involve knowledge of the anatomical structure of the human body, so there are far fewer barriers to the development of interaction, even to the extent of extending a new discipline, sports medicine and health promotion [10]. However, due to its high level of specialisation and practical threshold, although acceptance is increasing, the pace is indeed slow in relation to the public. Please note that (“the combination of sport and medicine” or “sport and medicine”) is a more popular term in China, so if you see the word and have difficulty understanding it, please read it as “sports medicine”.

The rapid development of Internet-based self-publishing platforms (Douyin, Bili Bili, Zhihu, etc.) has seen considerable development in recent years and has led to a new type of communication—new media communication. This type of mobile communication, based on smartphones and other mobile terminals, has lowered the threshold of access to health-related knowledge and made access to information faster, more convenient, freer and more accurate. For example, in the early stages of the COVID-19 pandemic, there was no public consensus on the characteristics of the COVID-19 pandemic and its protective measures. The public’s lack of understanding of the neoplastic pneumonia virus caused panic in the early stages of the pandemic. However, due to the existence of a large number of users on the self-published media platform, the official media and experts were able to disseminate knowledge to the public through this time-independent channel, and the initial users who received the information spread it to those around them. In addition, the existence of new media platforms was a major factor in the development of the pandemic. In addition, the existence of new media platforms has had a positive effect on reducing the cost of acquiring knowledge (buying books, going to medical institutions, etc.), which to a certain extent contributes to the efficient use of social resources and plays a role in economic development [11,12,13,14].

As of 2017, the number of Internet users in China reached 772 million, and this number is still growing. The number of Internet users in China is characterised by “more in the east and less in the west, more in the city and less in the countryside, and fewer young users than old ones” [15].

In the context of the normalisation of the pandemic, a series of reactions have occurred between sports medicine and media science; health knowledge related to sports medicine has caught the fast track of new media development, and the combination of the two has enabled it to spread rapidly among the community. More and more people are being exposed to sports health promotion through the new media and are actively participating in it [16,17].

The SWOT model is a qualitative analysis tool that analyses the four dimensions of internal strengths, internal weaknesses, external opportunities and external threats in the development process to provide guidance and avoid negative impacts; Strengths, Weaknesses, Opportunities and Threats (SWOT) analysis does not provide an analytical means to determine the importance of the identified factors or the ability to assess decision alternatives according to these factors. Although the analysis successfully pinpoints the factors, individual factors are usually described briefly and very generally. For this reason, SWOT analysis possesses deficiencies in the measurement and evaluation steps. Although the analytic hierarchy process (AHP) technique removes these deficiencies, it does not allow for measurement of the possible dependencies among the factors [18,19].

Meanwhile, the AHP model is a multi-criteria decision analysis (MCDA) method that helps to solve complex decision problems by constructing a decision factor structure for the problem to be analysed and assigning importance to each of these factors, making a distinction between the relative importance of each factor [20,21].

So, we have chosen a hybrid SWOT-AHP model to analyse the above issues, the SWOT-AHP model is an analytical tool that mixes the qualitative SWOT model with the quantitative AHP model to build a comparison matrix of the subfactors of the four factors S, W, O and T [22]. The relative importance of each subfactor and the strength of its influence on the development process is calculated in order to analyse more scientifically and accurately the impact of the combination of different factors on the overall development process [23]. The purpose of this study is to use SWOT to calculate the impact of the factors on the overall development process [24,25].

The aim of this study is to use the SWOT-AHP hybrid model to qualitatively analyse multiple aspects of new media empowered public health, coupled with quantitative data analysis, to provide a more scientific and sustainable development strategy.

## 2. Materials and Methods

### 2.1. Generation of the Factors of the SWOT Model

The various factors in the SWOT were generated by the joint consultation of 10 experts (researchers, academics and practitioners from hospitals, universities and new media platforms with in-depth knowledge of the development of the subject area) and the authors to analyse and judge the internal strengths [26], internal weaknesses, external opportunities and external threats of using the power of the media to promote the development of the concept of public health in sport in the context of the great development of the Internet. The Delphi method is a questionnaire method in which the questionnaire builder and the experts in the field work together over several rounds of discussion and deliberation to arrive at answers with relatively consistent opinions. The questionnaire used in this paper is a combination of the Delphi method and hierarchical analysis, so the questionnaire is administered by scoring the strategic intensity and relative importance of the issues to be discussed. Each factor of the SWOT to be discussed was first designed as a scale and discussed with 10 experts in several rounds. The questionnaire distributor considered the opinions returned by the experts during the discussion process and retained, deleted or optimised each sub-factor proposed until the experts’ opinions were unanimous when the sub-factors of the four SWOT factors were finally determined, as shown in Appendix A. Based on the results of the above scale, the newly designed questionnaire was then distributed to the 10 experts, who scored the relative importance and strategic strength of the results obtained, and after several rounds of discussion, a consensus was reached, as shown in Appendix A. Finally, the arithmetic mean of the scores obtained by the experts was calculated to construct a judgement matrix [27]. The specific process is shown in Figure 1.

### 2.2. SWOT-AHP Analysis

The SWOT model is a qualitative analysis model, and in order to quantitatively analyse the subfactors that have been selected, as shown in Figure 2, the AHP (hierarchical analysis) was used for calculation and analysis [28]. In the constructed AHP hierarchy, the interviewee was asked to compare the different sub-factors of the four SWOT factors on a two-by-two basis, according to their own ideas and to score them according to the AHP scale (Table 1) [20] in order to assess their importance. On the basis of the AHP scale, the interviewee was also asked to rate the expected strength of each sub-factor for subsequent calculations, with positive values for S, O and negative values for W, T. The higher the absolute value of the score, the greater the expected strength [29,30].

### 2.3. Calculation of the Weights of the Sub-Factors and Consistency Test of the Comparison (Judgment) Matrix

As everyone’s preferences are different, the scores given for the two comparisons of the sub-factors are not the same. The experts’ opinions were fed back through three rounds of information by the Delphi method, and a judgment (comparison) matrix was created to assess their consistency [31,32,33,34]. Take the example of internal strengths S.

Construct a comparison (judgment) matrix A, as shown in Table 2. ijerph-19-12813-t002_Table 2Table 2Comparison matrix A.*S**S*_1_*S*_2_*S*_3_*S*_4_S1a1b1c1d1S2a2b2c2d2S3a3b3c3d3S4a4b4c4d4Calculate the arithmetic sum of the subfactors in the judgment (comparison) matrix using the arithmetic mean method (sum product method), as shown in Table 3. ijerph-19-12813-t003_Table 3Table 3Comparison matrix A.*S**S*_1_*S*_2_*S*_3_*S*_4_S1a1b1c1d1S2a2b2c2d2S3a3b3c3d3S4a4b4c4d4SUMa1+a2+a3+a4b1+b2+b3+b4c1+c2+c3+c4d1+d2+d3+d4The above judgment matrix (comparison) array for normalization by column, that is, the value of each cell in the same column divided by the sum of all cells in the column to build a new normalized judgment matrix, as shown in Table 4. ijerph-19-12813-t004_Table 4Table 4Normalized judgment matrix A′.*S**S*_1_*S*_2_*S*_3_*S*_4_S1a1SUMb1SUMc1SUMd1SUMS2a2SUMb2SUMc2SUMd2SUMS3a3SUMb3SUMc3SUMd3SUMS4a4SUMb4SUMc4SUMd4SUMSUMa1+a2+a3+a4b1+b2+b3+b4c1+c2+c3+c4d1+d2+d3+d4Calculate the eigenvectors W (or weights, relative importance) for each subfactor.The eigenvector W of each row of the above established judgment matrix is calculated as the average of the normalized subfactors in each row of the matrix, calculated as: (1)W=∑i=1nsin=s1+s2+s3+…+snnThe resulting judgement matrix after calculation is shown in Table 5.ijerph-19-12813-t005_Table 5Table 5Judgement matrix after calculation.*S**S*_1_*S*_2_*S*_3_*S*_4_*W*S1a1SUMb1SUMc1SUMd1SUMW1S2a2SUMb2SUMc2SUMd2SUMW2S3a3SUMb3SUMc3SUMd3SUMW3S4a4SUMb4SUMc4SUMd4SUMW4Calculating *AW*.
(2)∑ij=1nSi×Wi(i=1,2,…,n),(j=1,2,…,n)The resulting judgement matrix after calculation is shown in Table 6.ijerph-19-12813-t006_Table 6Table 6Judgement matrix after calculation.*S**S*_2_*S*_3_*S*_4_*S*_5_*W**AW**S*_1_a1SUMb1SUMc1SUMd1SUMW1a1SUM×W1+b1SUM×W2+c1SUM×W3+d1SUM×W4*S*_2_a2SUMb2SUMc2SUMd2SUMW2a2SUM×W1+b2SUM×W2+c2SUM×W3+d2SUM×W4*S*_3_a3SUMb3SUMc3SUMd3SUMW3a3SUM×W1+b3SUM×W2+c3SUM×W3+d3SUM×W4*S*_4_a4SUMb4SUMc4SUMd4SUMW4a4SUM×W1+b4SUM×W2+c4SUM×W3+d4SUM×W4Calculate the maximum eigenvalue of the judgment matrix (λmax). (3)λmax=∑i=1nAWinwiCalculating the consistency index (CI).
(4)CI=λmax−nn−1Find the consistency test RI table (average random consistency index) [35] and select the appropriate values based on the number of sub-factors, as shown in Table 7.ijerph-19-12813-t007_Table 7Table 7Average random consistency index.*n*123456789RI000.520.891.121.261.361.411.46According to the formula CR = CI/RI, the CR value is calculated, and the consistency test of the judgment matrix is considered to be passed if CR < 0.1, and the consistency test of the judgment matrix is considered to be failed if CR > 0.1.

### 2.4. Calculating the Total Strength of the Factors in the SWOT Model and Constructing the SWOT Strategic Quadrilateral

Based on Chen’s research, it is concluded that subfactor strength = estimated strength of the subfactor x relative importance of the subfactor.
(5)$$\begin{array}{l} S_{i}=I_{i}\times P_{i}\left(i=1,2,\ldots ,nS\right)\\ W_{j}=I_{j}\times P_{j}\left(j=1,2,\ldots ,nW\right)\\ O_{k}=I_{k}\times P_{k}\left(k=1,2,\ldots ,nO\right)\\ S_{l}=I_{l}\times P_{l}\left(l=1,2,\ldots ,nT\right) \end{array}$$
where $S_{i}$ denotes the strength of each strength factor, $W_{j}$ denotes the strength of each weakness factor, $O_{k}$ denotes the strength of each opportunity factor and $T_{i}$ denotes the strength of each threat factor.

I and P denote the estimated strength and relative importance of each factor, X relative importance is the weight W (eigenvector) of each sub-factor in the judgement matrix, and estimated strength is the expected possible strength of the sub-factor given by the expert at the scoring stage.

For the total strength *S*′, total weakness *W*′, total opportunity *O*′ and total threat *T*′, the formula is
(6)$$\begin{array}{l} S^{\prime }=\sum_{i=1}^{ns}S_{i}\left(i=1,2,\ldots ,nS\right)\\ W^{\prime }=\sum_{j=1}^{nw}W_{j}\left(j=1,2,\ldots ,nW\right)\\ O^{\prime }=\sum_{k=1}^{no}O_{k}\left(k=1,2,\ldots ,nO\right)\\ T^{\prime }=\sum_{l=1}^{nt}T_{l}\left(l=1,2,\ldots ,nT\right) \end{array}$$

The calculated total strength *S*′, total weakness *W*′, total opportunity *O*′ and total threat *T*′ are brought into the Cartesian coordinate system and the four points are connected to each other to obtain the SWOT strategy quadrilateral.

### 2.5. Calculating the Strategy Vector $\left(\theta ,\rho \right)$

In the calculated SWOT strategy quadrilateral, the development strategy is determined according to the centre of gravity of the quadrilateral. At this point, we introduce the “strategic azimuth” to determine the positioning of the development strategy in more specific detail. The four quadrants of the coordinate system are defined as the pioneering aggressive area, the striving strategic area, the conservative strategic area and the resistant strategic area, and the area in each quadrant is then divided into two and defined as different small areas, as shown in the diagram; the area where the strategic azimuth is located, for the development of the general area, should follow.

When the strategic azimuth is in the first quadrant, the strengths and opportunities for development are relatively large and should be actively explored, either through strength or opportunity.

When the strategic azimuth is in the second quadrant, the strengths are weak but the opportunities are large and should be actively exploited.

When the strategic azimuth is in the third quadrant, you are at a disadvantage and a threat, so you should be conservative and avoid them.

When you are in the fourth quadrant, you have great strengths but also great threats, so you should make the most of your strengths and complement your weaknesses.

The quadrants in the coordinate system corresponding to the specific partitions are shown in Table 8.

Determine the strategic azimuth based on the coordinates of the centre of gravity of the strategic quadrilateral θ.
(7)$$P=\left(X,Y\right)=P\left(\frac{\sum xi}{4},\frac{\sum yi}{4}\right)$$Using inverse trigonometric functions to calculate θ.
(8)$$\theta =\arctan\frac{Y}{X}$$Determine strategic intensity factors based on positive and negative strategic intensities ρ.

At this point, we introduce two concepts of strategic positive intensity (*U*) and strategic negative intensity (*V*).

Positive strategic intensity is the result of the common action of external opportunities and intrinsic strengths of development. If the intrinsic strengths remain unchanged, changes in external opportunities will cause changes in the positive intensity of development strategies, calculated by the formula.
(9)$$U=O^{\prime }\times S^{\prime }$$

Negative strategic intensity is the result of the simultaneous action of external threats to development and intrinsic disadvantage. If intrinsic disadvantage remains unchanged, changes in external threats cause changes in the negative intensity of development strategies, calculated as follows.
(10)$$V=T^{\prime }\times W^{\prime }$$

Once the type of development has been determined based on the strategic azimuth, it is also important to determine what intensity of development is more appropriate, based on the combination of positive and negative strategic intensities. A decreasing strategic intensity coefficient indicates a more conservative intensity of development, an increasing strategic intensity coefficient indicates a more aggressive intensity of development, and if the strategic intensity coefficient is in the middle of the range, the current intensity of development should be maintained, steadily moving forward and adjusted according to the actual situation.

The strategic intensity factor ρ is calculated as:(11)$$\rho =\frac{U}{U+V}$$

## 3. Results

### 3.1. Each Factor in the SWOT Model

#### 3.1.1. Strength (*S*)

*S*_1_ New media is not affected by time and space in the dissemination of health concepts and health knowledge.

In the past, the public’s access to health-related knowledge and ideas was largely limited by time and space [36], for example, by the need to wait for a specific broadcast time for television programmes or to travel to a specific location for lectures. However, with the advent of new media, the public’s space and time to receive relevant information has been freed up to take advantage of the fragmented time to receive the information they want to know. The high degree of freedom, strong accuracy and wide accessibility are the greatest advantages of new media in disseminating relevant content [37]. The public does not need to devote a large chunk of time to learning about the content, but through the new media, the public health concept is further established to achieve a subtle effect.

*S*_2_ The variety of content presentation

Some of the content of traditional communication methods is boring to the young and middle-aged population, and the time costs involved are unacceptable to young and middle-aged people who live a fast-paced life and have heavy workloads, and the content is highly specialised and requires a lot of time to read and study, which makes it less acceptable. However, due to the huge number of young and middle-aged people, there is a huge sub-health group, with cervical spondylosis [38], obesity [39] and the gradual increase in the three high symptoms (hypertension, hyperlipidaemia and hyperglycaemia) [40] being neglected by patients. The development of new media platforms has made health-related content available in a form more suitable for the young and middle-aged population, and through live webcasts, illustrations and texts, in a humorous way, the young and middle-aged population can learn about it in an easy-to-understand and fun way in less time [37].

*S*_3_ The emergence of new media has increased the flexibility of the public’s identity in the chain of communication of health content.

As a mainstream social platform in contemporary society, new media not only provides a link between the supply and demand of health knowledge distributors and audiences, but also creates a communication hub that is interconnected and breaks through space and time [41]. Through this communication hub, people from different regions can communicate and discuss their experiences, and when exercising, they can clock in and motivate each other via the Internet, which increases the motivation and acceptability threshold for exercise while satisfying their personal wishes. At the same time, the audience of the content can, to a certain extent, transform themselves into the disseminators of the relevant content through the platform, extending the nodes of content dissemination, which, with the accumulation of huge user data, become unmissable one by one.

*S*_4_ New media platforms are more accurately “user profiling” through algorithmic mechanisms.

Compared to broad communication, the needs of each audience are not the same, and the new media platform, through the algorithm mechanism, accurately responds to the number of times the user uses a certain part of the content of the user profile. This forms a “virtual profile” for different audiences; the most relevant content is pushed to take into account the core audience of the content. The most relevant content is delivered by taking into account the core content, the audience’s needs, the audience’s experience and other issues, as well as the precise information that the audience needs most from their perspective. By comparing the content with the audience, the audience will have a more precise understanding of what they need to know and the form of practice they need to carry out [42].

#### 3.1.2. Weakness (*W*)

*W*_1_ The lack of a pre-qualification system and subsequent content evaluation system on media platforms has led to a weak sense of responsibility among media professionals.

Compared to traditional media, such as television, newspapers and news, new media lacks an effective ‘gatekeeper’ system [43], where content is published only for sensitive issues, and the content reviewers on the platforms are not health professionals and are not effective in pre-screening content. The low vetting threshold makes some media professionals no longer start with the value of integrating sports and medicine to promote health, but rather sacrifice media credibility to gain traffic and short-sighted interests, losing the original intention of using new media as a “fast track” to spread the idea of integrating sports and medicine to meet the public’s most basic fitness and recreation needs. This is the reason for the loss of the original intention of using new media as a “fast track” to disseminate the concept of physical medicine to meet the basic needs of the public. At the same time, compared to other activities in public life, participation in physical health activities needs to be carried out with the knowledge of professionals in order to enhance safety. It increases the cost of time spent by readers in screening the reliability of the content but also adversely affects the public who choose the wrong content for their physical activities, with negative consequences [44].

*W*_2_ Different media literacy of new media audiences.

Media literacy refers to people’s ability to choose, understand, question, evaluate, create and produce and respond thoughtfully to the various messages conveyed by the media, and is related to the age, education and life experience of the audience, with age being the more prominent issue that leads to differences in audience media literacy [45]. Content creators are more concentrated in the youth and middle-aged groups, while older groups are less competent in the use of new media. The rich and varied processes designed by new media creators are redundant for most older people, and the content is less focused on what older people need, such as specific diseases, thus ignoring the special needs of older people to a certain extent. The load and movements are not in line with the physiological characteristics of the elderly. Older people are already less able to choose and use new media, and this, combined with the fact that content creation is more focused on young and middle-aged people, leaves the disadvantaged groups of older people at a loss when it comes to choosing content. These problems will become more acute as the ageing process intensifies [46].

*W*_3_ Weak protection of personal information of new media users and spam reduces public perception of new media.

The public who are willing to use new media or Internet+ will actively search for the information they want to get, but as each person has different purposes and needs to conduct personalised searches on new media platforms, based on the characteristics of Internet platforms, the public will inevitably leave behind certain personal privacy data. However, limited by the imperfection of the relevant system and the lack of regulatory measures, most people will receive spam messages after leaving personal information. The weakness of personal information protection is an issue that needs to be addressed by the entire Internet and new media industry [47].

#### 3.1.3. Opportunity (*O*)

*O*_1_ National-level policies to help the development of new media and the integration of sports and medicine to promote health are both on the fast track.

In June 2016 and April 2018, the General Office of the State Council issued the “Guidance on Promoting and Regulating the Development of Health Care Big Data Applications” and the “Opinions on Promoting the Development of “Internet + Medical Health” to promote the construction of health information for all. The concept of health and a healthy lifestyle is an important part of the health of the whole population, while, for the Internet, as early as in 2015, the State Council has already issued the “Internet +”, which is the deep integration of the Internet with all areas of the economy and society [48,49], to promote technological progress and organisational change, enhance the innovation and productivity of the real economy and form a wider range of economic activities with the Internet as the infrastructure and innovation factor. The Internet is a new form of economic and social development that uses the Internet as an infrastructure and innovation factor. The promotion at the national level is extremely important for the development of the Internet and health concepts. The State Council’s emphasis on the importance of the Internet + health in two separate occasions is profound proof of the state’s need for the dissemination of health concepts and lifestyles on the Internet.

*O*_2_ Active academic discussion on the integration of sports medicine for the interaction between health and new media.

In contrast to the previously fragmented development of the various disciplines, there has been considerable academic discussion and debate in recent years on the integration of sports, medicine and media, with the interpenetration of the disciplines drawing on each other to promote public health, which is no longer solely the responsibility of medicine, but also emphasises the active role of sport as a means of promoting public health and preventing chronic diseases. The public is increasingly willing to accept sport as a complementary means of disease prevention and to embrace new media as a form of communication, willing to devote fragmented time to learning about it and applying it to their lives on new media platforms, and sport is increasingly seen as an important part of the public’s free time [50,51]. According to relevant reports, the scale of active users of sports and fitness apps rose rapidly to 89.28 million during the pandemic, nearly doubling year on year [52].

*O*_3_ COVID-19 pandemic is a special time when the public’s health aspirations have increased and new media are being used more frequently.

In the fight against the COVID-19 pandemic, there is a general consensus on the importance of good physical fitness [53]. All things being equal, the susceptible population for the COVID-19 pandemic is concentrated in the elderly and the sub-healthy, and this population is characterised by a weaker physical condition. According to research, people who are in better health have a lower rate of infection and a faster recovery rate from the disease [54,55]. The example of the Newcastle pneumonia outbreak shows the role of good health in the early stages of a public health emergency as an outpost for disease prevention and highlights the important role of good health and a healthy lifestyle in reducing the prevalence of more diseases. The need for health is not just a matter of increased awareness, but also a deepening of real needs. The “exercise card” model of apps, such as Keep, and the “online coaching” model of apps, such as Tik Tok, have created new forms of physical activity—relying on new media platforms. Home exercise on new media platforms not only alleviates the need for exercise and health demands of people at this particular time but also provides a new type of exercise solution for the majority of people who do not have much free time to go to sports activities in the post-pandemic period [56].

*O*_4_ Rapid development of online platforms gives ground to the prospect that new media can empower healthiness.

As the country with the highest usage and penetration of online platforms, China leads the world in the field of online mobile payments. The popularity of mobile payments has brought about not only a revolution in payment methods, but also a significant increase in the usage of the corresponding platforms, relying on which the prerequisites for new media to promote healthy development can be met—large amounts of user data. For example, Alipay, a platform specialising in cloud payments, is building an online healthcare system; WeChat, a real-time communication platform, is building a system for sharing and disseminating sports and health knowledge and feedback; and Jitterbug, a short video entertainment platform, is recruiting more professionals and building a more accessible collection of health-related content. These platforms are also innovating new technologies in the process of development, such as immersive human–computer interaction experience systems, based on 5G. New technologies will enrich the new experiences people have when performing sports in order to increase users’ enjoyment of using them.

#### 3.1.4. Threats (*T*)

*T*_1_ The intensification of malicious health marketing caused by the excessive injection of commercial capital.

The rapid development of new media platforms has not only promoted public awareness of health and increased the initiative to engage in physical activities but also promoted the consumption of various health-related products. The media platforms are also full of publicity, exaggerating and confusing the public with health care claims [57]. Some organisations even exaggerate the risks of certain diseases based on a lack of information and take advantage of the public’s lack of knowledge and fear of diseases to insert relevant products into their content, greatly diluting the public service of promoting health through the integration of body and medicine through new media, making the whole field full of commercialism. In the field of physical health, the drive for profit has made the public bear the ill effects of the commercial war between capitals [58].

*T*_2_ Uncoordinated and uneven regional development exacerbates the health perception divide.

With the popularity of mobile phones, a mobile terminal, the way in which the public receives information has become more convenient and diversified, breaking through the limitations of time and space, which is the greatest advantage of new media in differentiating them from traditional media, and to a certain extent filling the information gap caused by the imbalance in regional development [59]. The improved health-related perceptions of the masses and the increased self-awareness of physical activity brought about by the new media require a realistic ground for the masses to facilitate their practice. However, there is a regional skew in resources, as advanced medical tools, fitness equipment and wellness concepts are the first to reach the better developed regions, while people in areas where health resources are scarce are unable to find local conditions to practise the content they receive through self-media. As the content becomes available on a larger scale to people in better developed areas, the content creators, in order to improve the quality of the content they create, will focus on the needs of the developed areas and update the content to promote their further development, further widening the gap between the regions and discouraging people in areas where health resources are scarce from participating in this new media-enabled cycle of health integration.

*T*_3_ New media and hospitals as nodal inversions in the great circle of health promotion through the integration of physical medicine.

New media, as a tool to empower the integration of physical medicine for health development, is more about spreading knowledge to prevent diseases than relying on the knowledge disseminated by new media to treat diseases. The content of new media-based communication has a certain outpost role in promoting public health. Through the dissemination of relevant knowledge and other means, the public’s concept of healthy living is improved, health awareness is cultivated and the public’s enthusiasm to participate in the general cycle of integration of body and medicine for health development is enhanced [60]. However, it is putting the cart before the horse if the public treats the new media as a one-to-one professional doctor, relying on cultural communication content to treat illnesses while ignoring the role of hospitals.

### 3.2. SWOT-AHP Model

The judgment (comparison) matrices in the SWOT-AHP model, the weights (relative importance) of the factors calculated using arithmetic averaging, *AW*, the maximum eigenvectors ($\lambda_{\max}$) and the CI and CR values used for the consistency test of the judgment matrices are shown in the table; the consistency test CR values of the judgments constructed for the four SWOT factors are all less than 0.1; therefore, all judgment matrices are considered to be passed. The details are shown in Table 9.

Strategic strength of each sub-factor and total strength in the SWOT-AHP model. The details as shown in Table 10.

### 3.3. Construction of the Strategic Quadrilateral and the Strategic Vector $\left(\theta ,\rho \right)$

The coordinates of the centre of gravity of the strategic quadrilateral are P(X, Y) = (0.21075, −0.0496)

Strategic azimuths is $\theta =\arctan\left(\frac{-0.0496}{0.21075}\right)=-13.243^{\circ }$

The strategic positive vector is *U* = 22.1194

The strategic negative vector is *V* = 19.0724

The strategic intensity factor is ρ=0.53699

The SWOT strategy quadrilateral is shown in Figure 3.

Substitute the strategic vector angles into the coordinate system, as shown in Figure 4.

## 4. Discussion

Based on the SWOT-AHP empirical analysis, the following five points should be prioritised as solutions.

Firstly, strengthening health-related social ethics and promoting the prevalence of healthy living. As the third industrial revolution progresses, new media has become an important way for the public to access health information, seek health assistance, carry out recreational activities and carry out medical services. The quality of information plays a crucial role in the dissemination of information, and the construction of a corresponding sense of public order and moral identity requires the joint efforts of the government, society, new media platforms, content creators and content audiences. At the government level, relevant laws and regulations should be improved as soon as possible, and an effective legal accountability system should be established, so that laws can be followed and violations will be investigated [61]; the platform should clearly supervise and manage responsibilities, establish an efficient and rigorous system for dealing with violations and restrict platform access or even ban accounts for content creators who violate public order and morality, so as to purify the platform environment; content creators should clearly ensure that the content is true and in the public interest. The red line is that content creators should be clear about the truthfulness of their content and the public good [62] and should create content that is socially acceptable based on their own experience and knowledge, the real characteristics of the disease and the allocation of resources in the region. The laws and regulations, the rules of the platform and the rigour of the creators will allow the information to be disseminated under the premise of social order and morality, and the new media platform will play a role in helping to promote the concept of public health and the active participation of the public in sports and health activities, using scientific and high-quality content as a guide and forming a trend of health for all in society.

Secondly, enhancing the accuracy of content delivery and ensuring that content creation is fun. The uncoordinated regional development and the age-generation gap between people have reduced the universality of the content pushed out, and the platform should continue to optimize the establishment of a more accurate “user profile” and push out relevant content based on the search volume and attention of users. However, the “user profile” created by the platform should be strictly adhered to for the platform’s use only, strictly guarding key information, such as user preferences and protecting users’ personal privacy. Content creators should first consider the population and regional characteristics of the content at the point of creation. In terms of content, it should be clear for which users the content is recommended to be more suitable, to avoid ambiguity in content creation. In addition, the platform should also establish a “health information” platform. In addition, platforms should establish an “anti-information cocoon” mechanism to avoid the fixation and homogenisation of information received by users as a result of user precision. Government media and mainstream health-related media should encourage more official accounts to join the platform, taking into account the realities of regional development and creating content suitable for the local population, in keeping with the public interest nature of the content. Content creators should also consider reducing the jargon of the language as much as possible while ensuring scientific and rigorousness, in the form of easy-to-understand language expressions.

Thirdly, platforms strengthen the pre-qualification system for content creators and the post-qualification system for content regulation. Due to the special nature of sports and health-related content, unprofessional content can lead to poor user experience and even harm to the body. Therefore, the platform should recruit a certain number of professional composite talents as content creators’ qualification auditors and set up a certain threshold audit system before the creators are admitted to the platform, so that only those who pass the audit can be admitted to the platform and obtain official certification. After the creator is on the platform, the auditors should regularly review the content published by the creator to review the professionalism of the content and try to avoid the dissemination of wrong knowledge to avoid misleading the recipients of the content. However, at the same time, for non-professional creators who have already published other content and later want to transition to publishing health-related content, they should be given certain professional knowledge training and be audited according to their specific situation. A ‘one-size-fits-all’ type of auditing system will discourage people who want to extend their content dissemination nodes, transforming the masses into communicators, and is not in line with the open and inclusive nature of new media.

Fourth, keeping health communication clean and combating ethical business malpractice. Apart from being a platform for information sharing, the new media platform is also a business platform. The explosion of the e-commerce economy is the result of the enhancement of the national consumption concept and consumption ability. However, as the relevant laws and regulations are in the process of being improved, the lack of regulation, the incomprehensiveness of the regulatory dimension and the disorder of the regulatory process have given an opportunity to a part of the merchants who do have commercial ethics to take advantage of. The whole society’s health contains many aspects, and the control of health products is the most important issue that needs attention. The platform should play its role as a regulator and combat discrimination and stigmatisation, general entertainment, over-packaging and misrepresentation of health-related work with the utmost severity, in order to prevent the over-profiting of health-related products. In addition, the platform should establish a real feedback rating system for users, inviting them to evaluate the health products they buy on the platform while ensuring the quality of the service and to rate the products sold based on their opinions. A standardised industry chain will be established, with the platform, content producers and content recipients, participating in the construction, supervision and management of the platform.

Finally, making the most of its strengths to raise the community’s awareness of health. We will continue to make use of information technology to empower innovative development. We will continue to take advantage of the high degree of freedom, interoperability and accuracy of self-published media to reinforce the new media’s ability to promote health development through the integration of sports and medicine, so that more people can participate in the cycle, experience the benefits and pleasures of participating in the cycle, build a solid base of participants, increase the number of participants and raise the common understanding of health in society as a whole.

Public health encompasses many aspects; the increase in awareness of healthy living, the development of behavioural practices that promote physical health, the increase in sports participation and the orderly and controlled development of public health-related industries all reflect the building and improvement of public health in a region or country. The above perspectives involve macro control by the government and the platform and micro but extensive participation by the public. The three are interrelated as a whole and a problem in any one link will hinder the overall development. According to the SWOT-AHP model, it can be seen that there are both strengths and threats to China’s use of new media for public health. The huge internal strengths are not an excuse to ignore the threats, nor are the complex external threats simply a stumbling block to development; we should use the threats as a bottom line and maximise the strengths above the bottom line to expand the upper limit of development. Take advantage of the favourable policy and social environment, guard the red line of platform content regulation and promote public health to ride on the new media fast track to sustainable development by taking public demands as a grip.

It should also be noted that there is no “sequential relationship” between the two dimensions of health literacy and physical health, i.e., the two are mutually reinforcing and integrated, and it is inaccurate to assume that either is more important than the other. It is inaccurate to assume that the two are more important than each other. At different times, they are both primary and secondary aspects. Secondly, the role of new media in the promotion of public health is more of a tool, but it is the public itself that plays the decisive role. In the process of development, the role of new media should not be exaggerated to the neglect of the public, as the main factor in the promotion of public health.

In addition, there may be some shortcomings in this study, for example, the author may mix some subjective factors in the process of investigating and taking evidence on various factors of SWOT, which is unavoidable in the SWOT-AHP model. It is hoped that more analytical methods can be applied in this field in the future to further improve the scientific nature of the study. For example, the author ignores the issue of “health literacy” in the construction of the SWOT model (health literacy is defined as the ability to access, understand, appraise and apply health information; merely having health information does not guarantee behaviour change).

Finally, we would like to point out that the author’s starting point is more about what is happening in Chinese society, as China is at the forefront of new media development and usage in the world and hopes that the development of the phenomenon studied in China will serve as a “template” for the rest of the world, helping to advise other countries or regions on the relevant developments that are taking place in order to help other countries and regions to avoid harm.

## 5. Conclusions

The integration of the three disciplines of media, sport and medicine is a golden opportunity to rapidly raise awareness, widely disseminate knowledge and increase participation in health campaigns for the whole population. This paper uses the SWOT-AHP model to conduct an empirical study on the development of new media-enabled sports and medical integration strategies for the promotion of health for all. The results show that under the four dimensions of the SWOT model, threats and strengths coexist and belong to a resistant development posture, and while there are huge advantages, there are also corresponding external threats, and there is huge room for improvement in development prospects. In the future, through a series of regulatory measures, laws and regulations in place; the regulation of health-related business ethics; and the continued optimisation of platforms and content creators, more and more people will participate in the great cycle of new media empowering the integration strategy of physical medicine to promote the health of all people, and it will become a new social trend.

## Figures and Tables

**Figure 1 ijerph-19-12813-f001:**
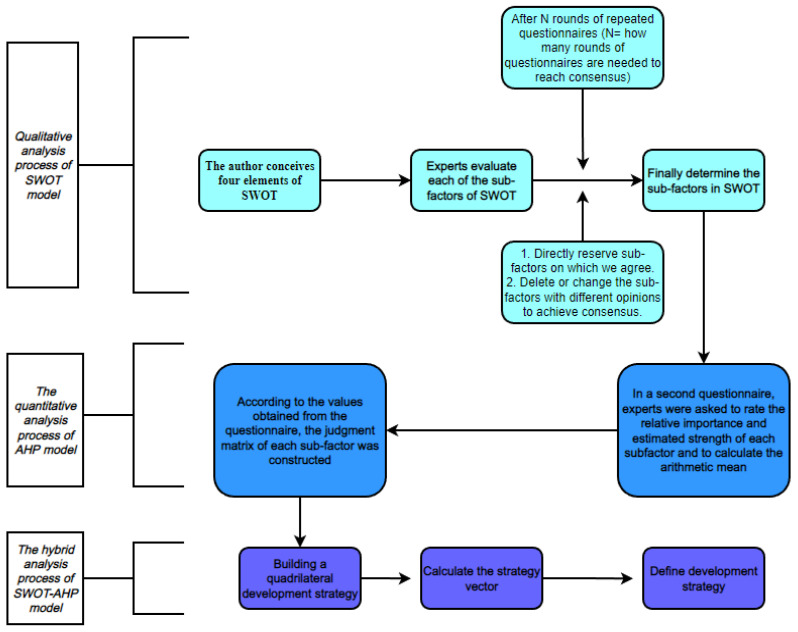
The progress of SWOT-AHP analysis.

**Figure 2 ijerph-19-12813-f002:**
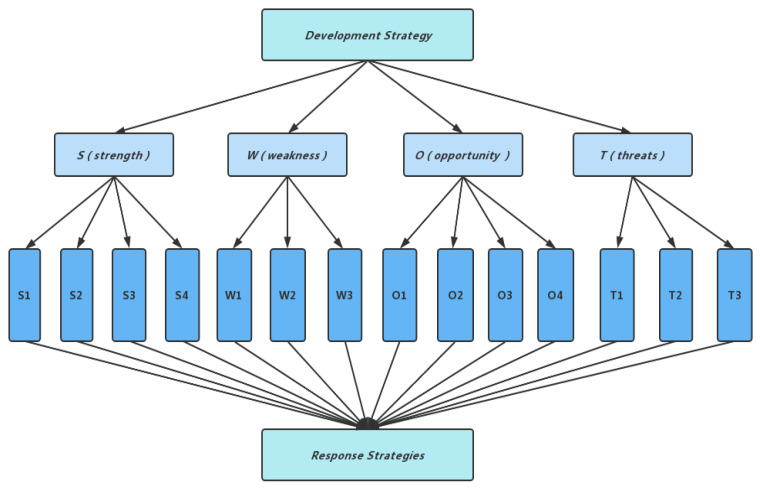
Hierarchical analysis structure graph.

**Figure 3 ijerph-19-12813-f003:**
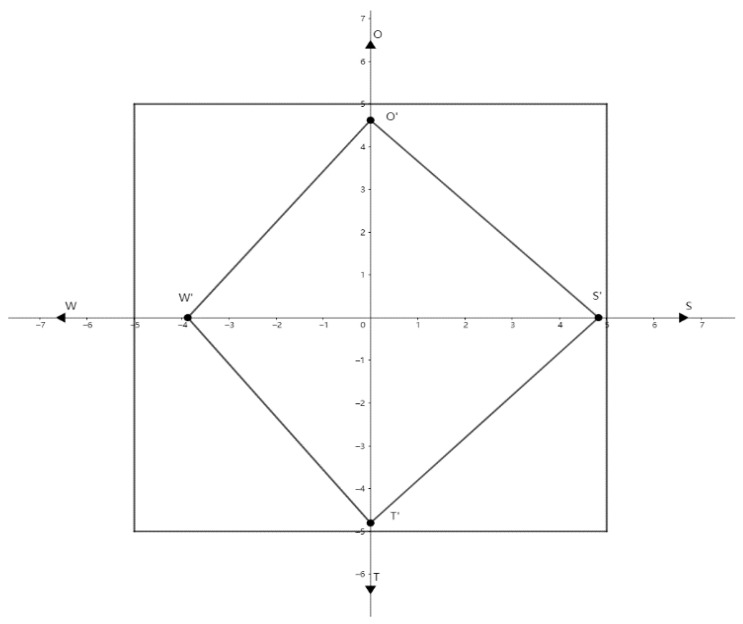
The SWOT strategy quadrilateral. *S*: Strength, *W*: Weakness, *T*: Threats and *O*: Opportunity.

**Figure 4 ijerph-19-12813-f004:**
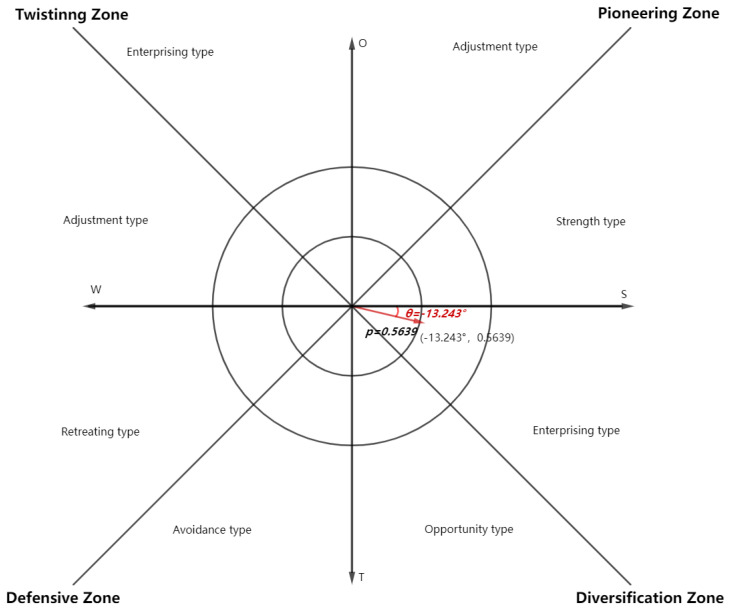
Development strategy vector and interval maps. *S*: Strength, *W*: Weakness, *T*: Threats and *O*: Opportunity.

**Table 1 ijerph-19-12813-t001:** AHP scale.

Intensity of Importance	Definition	Explanation
1	Equal Importance	Two activities contribute equally to the objective
3	Moderate importance	Experience and judgement slightly favour one activity over another
5	Strong importance	Experience and judgement strongly favour one activity over another
7	Very strong or demonstrated importance	An activity is favoured very strongly over another; its dominance is demonstrated in practice
9	Extreme importance	The evidence favouring one activity over another is of the highest possible order of affirmation
2, 4, 6, 8	Importance between the above levels	
1/2, 1/3, 1/4, 1/5, 1/6, 1/7, 1/8, 1/9	If sub-factor A is less important than sub-factor B, then intensity of importance of A is the inverse of intensity of importance of B	A reasonable assumption

**Table 8 ijerph-19-12813-t008:** Coordinate system and quadrant partitioning of the SWOT-AHP model.

First Quadrant		Second Quadrant		Third Quadrant		Fourth Quadrant	
Pioneering strategic areas		Ambitious strategic areas		Conservative strategic areas		Resistant strategic areas	
Type	Azimuth field	Type	Azimuth field	Type	Azimuth field	Type	Azimuth field
Strength type	$\left[0,\frac{\pi }{4}\right)$	Aggressive type	$\left[\frac{\pi }{2},\frac{3\pi }{4}\right)$	Retreating type	$\left[\pi ,\frac{5\pi }{4}\right)$	Adjustment type	$\left[\frac{3\pi }{2},\frac{7\pi }{4}\right)$
Opportunity Type	$\left[\frac{\pi }{4},\frac{\pi }{2}\right]$	Adjustment type	$\left[\frac{3\pi }{4},\pi \right)$	Avoidance type	$\left[\frac{5\pi }{4},\frac{3\pi }{2}\right)$	Aggressive type	$\left[\frac{7\pi }{4},2\pi \right)$

**Table 9 ijerph-19-12813-t009:** The details about the SWOT-AHP model.

SWOT-AHP Group	Comparison Matrices	SUM	Relative Importance (Weighting) W	*AW*	$\lambda_{\max}$	CI	CR
*S*	$\left\{\begin{array}{cccc} 1 & 1/2 & 1/3 & 1/7\\ 2 & 1 & 1/5 & 1/6\\ 3 & 5 & 1 & 1/4\\ 7 & 6 & 4 & 1 \end{array}\right\}$	$\left\{13\;12.5\;5.533\;1.560\right\}$	WS1 = 0.0672 WS2 = 0.0942 WS3 = 0.2429 WS4 = 0.5956	AWS1 = 0.2804 AWS2 = 0.3765 AWS3 = 1.0652 AWS4 = 2.6031	4.2301	0.0766	0.0772
*W*	$\left\{\begin{array}{ccc} 1 & 1/7 & 1/2\\ 7 & 1 & 4\\ 2 & 1/4 & 1 \end{array}\right\}$	$\left\{10\;1.39\;5.5\right\}$	WW1 = 0.9782 WW2 = 0.7151 WW3 = 0.1871	AWW1 = 0.2935 AWW2 = 2.1483 AWW3 = 0.5615	3.00198	0.0009	0.0018
*O*	$\left\{\begin{array}{cccc} 1 & 2 & 1/2 & 4\\ 1/2 & 1 & 1/3 & 3\\ 2 & 3 & 1 & 5\\ 1/4 & 1/3 & 1/5 & 1 \end{array}\right\}$	$\left\{3.75\;6.333\;2.033\;13\right\}$	WO1 = 0.2840 WO2 = 0.1715 WO3 = 0.4708 WO4 = 0.0736	AWO1 = 1.1569 AWO2 = 0.6914 AWO3 = 1.9217 AWO4 = 0.2959	4.05136	0.0171	0.0189
*T*	$\left\{\begin{array}{ccc} 1 & 2 & 1/3\\ 1/2 & 1 & 1/5\\ 3 & 5 & 1 \end{array}\right\}$	$\left\{4.5\;8\;1.533\right\}$	WT1 = 0.2298 WT2 = 0.1221 WT3 = 0.6521	AWT1 = 0.6902 AWT2 = 0.3667 AWT3 = 1.9484	3.00369	0.0018	0.0033

**Table 10 ijerph-19-12813-t010:** The strength of each sub-factor of the model.

SWOT Group	Relative Importance (Weighting) W	Estimated Intensity	The Strength of Each Sub-Factor	Total Strength
*S*	WS1 = 0.0672 WS2 = 0.0942 WS3 = 0.2429 WS4 = 0.5956	S1 = 1 S2 = 2 S3 = 4 S4 = 6	0.0671 0.1884 0.9716 3.5736	4.8007
*W*	WW1 = 0.9782 WW2 = 0.7151 WW3 = 0.1871	W1 = −2 W2 = −5 W3 = −1	−0.1956 −3.5750 −0.1871	−3.9577
*O*	WO1 = 0.2840 WO2 = 0.1715 WO3 = 0.4708 WO4 = 0.0736	O1 = 4 O2 = 3 O3 = 6 O4 = 2	1.1360 0.5145 2.8254 0.1472	4.6231
*T*	WT1 = 0.2298 WT2 = 0.1221 WT3 = 0.6521	T1 = −3 T2 = −2 T3 = −6	−0.6897 −0.2444 −3.8874	−4.8215

## Data Availability

The datasets used and analyzed during the current study are available from the corresponding author on reasonable request.

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
