# Peer review of "Where Is the Way Forward for New Media Empowering Public Health? Development Strategy Options Based on SWOT-AHP Model"

_ijerph, 2022, doi:10.3390/ijerph191912813_

Round 1

Reviewer 1 Report

Dear editors,

Thanks for the opportunity to review this article about development strategy options based on the SWOT-AHP model for new media empowering public health. I found the article and the research interesting, and I consider that they can be helpful for other researchers. However, some minor aspects could be commented on, trying to improve the overall readability of this good article. These are the following:

- In the Introduction section, perhaps the authors could slightly describe some aspects of using these interventions from the users' perspective. Many researchers do not often assess this perspective, which can help improve usability and lower attrition rates.

- Some ideas about the importance of economic aspects could also be mentioned, as they are undoubtedly crucial for developing interventions and evaluations.

- In the Methods section, the authors describe, «The various factors in the SWOT were generated by the joint consultation of 10 experts and the authors to analyse and judge the internal strengths». Potential selection bias is essential in this research. Therefore, I would invite the authors better to describe the experts' selection and recruitment process. It would also be helpful if they could assess, if possible, the potential selection bias when considering the external validity of their findings.

- In the Results section, there are several comments, even with references, that perhaps could fit better in the Discussion section, as they seem more like the authors' comments than the results obtained. As I understand that this kind of research could be difficult to separate results from the discussion, this is just a proposal.

- The Discussion section, otherwise, seems a bit short and with few references. Therefore, the comments and references in the results section could be moved to the discussion section. I believe that this could help to improve the readability of the article. These are just proposals.

- I would encourage the authors to describe the limitations of the research, like the potential selection bias, at the end of the discussion.

Author Response

Response to Reviewer 1 Comments

Point 1: - In the Introduction section, perhaps the authors could slightly describe some aspects of using these interventions from the users' perspective. Many researchers do not often assess this perspective, which can help improve usability and lower attrition rates.

- Some ideas about the importance of economic aspects could also be mentioned, as they are undoubtedly crucial for developing interventions and evaluations.

Response 1: For example, in the early stages of the COVID-19 epidemic, there was no public consensus on the characteristics of COVID-19 and its protective measures. The public's lack of understanding of the Neoplastic pneumonia virus caused panic in the early stages of the epidemic. However, due to the existence of a large number of users on the self-published media platform, the official media and experts were able to disseminate knowledge to the public through this time-independent channel, and the initial users who received the information spread it to those around them. In addition, the existence of new media platforms was a major factor in the development of the epidemic. In addition, the existence of new media platforms has had a positive effect on reducing the cost of acquiring knowledge (buying books, going to medical institutions, etc.), which to a certain extent contributes to the efficient use of social resources and plays a good role in economic development.

Point 2: - In the Methods section, the authors describe, The various factors in the SWOT were generated by the joint consultation of 10 experts and the authors to analyse and judge the internal strengths». Potential selection bias is essential in this research. Therefore, I would invite the authors better to describe the experts' selection and recruitment process. It would also be helpful if they could assess, if possible, the potential selection bias when considering the external validity of their findings.

Response 2: The Delphi method is a questionnaire method in which the questionnaire builder and the experts in the field work together over several rounds of discussion and deliberation to ar-rive at answers with relatively consistent opinions. The questionnaire used in this paper is a combination of the Delphi method and hierarchical analysis, so the questionnaire is administered by scoring the strategic intensity and relative importance of the issues to be discussed. Each factor of the SWOT to be discussed was first designed as a scale and dis-cussed with 10 experts in several rounds. The questionnaire distributor considered the opinions returned by the experts during the discussion process and retained, deleted or optimised each sub-factor proposed until the experts' opinions were unanimous when the sub-factors of the four SWOT factors were finally determined, as shown in Supplementary Table 1. Based on the results of the above scale, the newly designed questionnaire was then distributed to the 10 experts, who scored the relative importance and strategic strength of the results obtained, and after several rounds of discussion, a consensus was reached, as shown in Supplementary Table 2. Finally, the arithmetic mean of the scores obtained by the experts was calculated to construct a judgement matrix. The specific process is shown in Figure 1.

Point 3: - In the Methods section, the authors describe, «The various factors in the SWOT were generated by the joint consultation of 10 experts and the authors to analyse and judge the internal strengths». Potential selection bias is essential in this research. Therefore, I would invite the authors better to describe the experts' selection and recruitment process. It would also be helpful if they could assess, if possible, the potential selection bias when considering the external validity of their findings.

Response 3: Indeed, much of the results section and the discussion section are not easily described separately, and if they are described separately they may cause confusion to a proportion of readers who are less familiar with the SWOT-AHP model, so we have chosen to introduce more literature in the results section and the reference section to facilitate the reader's desire to explore further from the cited literature after reading this paper.

In addition, there may be some shortcomings in this study, for example, the author may mix some subjective factors in the process of investigating and taking evidence on various factors of SWOT, which is unavoidable in the SWOT-AHP model. It is hoped that more analytical methods can be applied in this field in the future to further improve the scientific nature of the study.

Supplementary Table 1.
Group
Sub-factors
Accept or Reject(√ OR ×)
Estimated intensity
Strengths
(S)
S1

1-5 (The higher the absolute value of the score, the more important it is)
S2

S3

S4

Opportunities(O)
O1

O2

O3

O4

Weakness
(W)
W1

(-1)-(-5) (The higher the absolute value of the score, the more important it is)
W2

W3

Threats
(T)
T1

T2

T3

Supplementary Table 2.
Using S group to explain.
Comparisons among subfactors within group S

Equal Importance
Moderate importance
Strong importance
Very strong or demonstrated importance
Extreme importance
Importance between the above levels
If sub-factor A is less important than sub-factor B, then Intensity of Importance of A is the inverse of Intensity of Importance of B

1
3
5
7
9
2,4,6,8
1/2,1/3,1/4,1/5,1/6,1/7,1/8,1/9

S1

S1
S1

S2
S1

S3
S1

S4
S2

S1
S2

S2
S2

S2
S2

S2
S3

S3
S3

S3
S3

S3
S3

S3
S4

S4
S4

S4
S4

S4
S4

S4

Reviewer 2 Report

The paper deals with a very important and innovative topic on the use of the hybrid SWOT-AHP model to qualitatively analyse multiple aspects of new media empowered exercise health promotion, coupled with quantitative data analysis, to provide a more scientific and sustainable development strategy.

The authors present a good introduction that well frames the problematic with a clear link to the objective of the study.

The title reflects the totality of the study and is very well elaborated in order to arouse the reader's interest.

Chapter 2. Materials and Methods is very exhaustive, but explanatory of the methodology used. The authors must explain the process of selection of the experts as well as the profile defined and intended for these experts.

Chapters 3. Results and 4. Discussion very objective and well developed.

Author Contributions is not filled in.

The set of bibliographical sources is adequate to the type of study and is scientifically up-to-date.

I congratulate the authors for the paper.

Author Response

Response to Reviewer 2 Comments

Point 1: Chapter 2. Materials and Methods is very exhaustive, but explanatory of the methodology used. The authors must explain the process of selection of the experts as well as the profile defined and intended for these experts.

Response 1: The Delphi method is a questionnaire method in which the questionnaire builder and the experts in the field work together over several rounds of discussion and deliberation to ar-rive at answers with relatively consistent opinions. The questionnaire used in this paper is a combination of the Delphi method and hierarchical analysis, so the questionnaire is administered by scoring the strategic intensity and relative importance of the issues to be discussed. Each factor of the SWOT to be discussed was first designed as a scale and dis-cussed with 10 experts in several rounds. The questionnaire distributor considered the opinions returned by the experts during the discussion process and retained, deleted or optimised each sub-factor proposed until the experts' opinions were unanimous when the sub-factors of the four SWOT factors were finally determined, as shown in Supplementary Table 1. Based on the results of the above scale, the newly designed questionnaire was then distributed to the 10 experts, who scored the relative importance and strategic strength of the results obtained, and after several rounds of discussion, a consensus was reached, as shown in Supplementary Table 2. Finally, the arithmetic mean of the scores obtained by the experts was calculated to construct a judgement matrix. The specific process is shown in Figure 1.

Supplementary Table 1.
Group
Sub-factors
Accept or Reject(√ OR ×)
Estimated intensity
Strengths
(S)
S1

1-5 (The higher the absolute value of the score, the more important it is)
S2

S3

S4

Opportunities(O)
O1

O2

O3

O4

Weakness
(W)
W1

(-1)-(-5) (The higher the absolute value of the score, the more important it is)
W2

W3

Threats
(T)
T1

T2

T3

Supplementary Table 2.
Using S group to explain.
Comparisons among subfactors within group S

Equal Importance
Moderate importance
Strong importance
Very strong or demonstrated importance
Extreme importance
Importance between the above levels
If sub-factor A is less important than sub-factor B, then Intensity of Importance of A is the inverse of Intensity of Importance of B

1
3
5
7
9
2,4,6,8
1/2,1/3,1/4,1/5,1/6,1/7,1/8,1/9

S1

S1
S1

S2
S1

S3
S1

S4
S2

S1
S2

S2
S2

S2
S2

S2
S3

S3
S3

S3
S3

S3
S3

S3
S4

S4
S4

S4
S4

S4
S4

S4

Reviewer 3 Report

Thank you for the opportunity to review this article, it deals with a very important current topic.

Different terms are used inconsistently and without definition throughout the study. The research attempts to deal with a rapidly developing service for the whole of China, although this is not evident initially. The paper is built on the assumption that health knowledge underpins good health. The lack of acknowledgment of the WHO’s work on the social and commercial determinants of health is a serious omission. Health education makes a contribution to health outcomes but only for those who can utilise it.  The paper mixes the concepts of health and wellbeing with illness prevention and accessing health services. The content is very broad and consequently superficial.    

Various terms are used inconsistently throughout the paper – eg sport, medicine, public health and promoting health, it would be helpful for the reader to understand the definitions used by the authors for these important constructs. In addition a new term kinesiology is introduced in the conclusion, which further confuses. The term health promotion is used interchangeably with health education – they are different – review

Discussion on the population demographics of China, numbers of people, age groups, social group, disability, regional variations and access to the internet via for example smart phones/other devices would add weight to the arguments. Also which age groups/ societal groups are more likely to access sports/ health devices/apps. Has there been any evidence to show that people are having health benefit from using the app/ devices etc. Using the term the public is too misleading when discussing a population of 1.4 billion people. A clearer understanding of which group of people the study is reviewing is required. There is also an assumption that people are accessing health information but this is not explored or confirmed with evidence.

L. 54 is chemical reactions the most appropriate phrase to use here?

L 60 – a critical discussion of SWOT would be useful early in the paper and a rationale for its choice.

L 80 – experts in what? Requires clarification.

L 258 – review use of word excellent – too emotive lacks supporting evidence

The work does not refer to the body of evidence on health literary

Generally throughout the work there are many unsupported sweeping statements  

What would regulation of the internet look like, how would it be achieved?

L 339 what percentage of the population using these apps/ devices engage in sport ? Clarity and objectivity required.

L344 what were the searches around covid protection how did they relate to general fitness – more clarity required.

The stages of the swot analysis could be clearer, the reader would benefit from supported guidance at each step/ figure.

The conclusion assumes that ‘improvements’ would continue. What guidance is required based on the research?

Where are the recommendations for the study?

Syntax and sentence clarity requires review.

Author Response

Response to Reviewer 3 Comments

Point 1: Different terms are used inconsistently and without definition throughout the study. The research attempts to deal with a rapidly developing service for the whole of China, although this is not evident initially. The paper is built on the assumption that health knowledge underpins good health. The lack of acknowledgment of the WHO’s work on the social and commercial determinants of health is a serious omission. Health education makes a contribution to health outcomes but only for those who can utilise it.  The paper mixes the concepts of health and wellbeing with illness prevention and accessing health services. The content is very broad and consequently superficial.    

Response 1:

We took this into consideration before conducting the study, but what we believe is that health knowledge and physical fitness are a mutually integrated developmental process. For example, in the preparation phase of primary and secondary school physical education classes, in the past, physical education teachers would often ask students to perform a circular rotation, but the knee joint does not have a "circular rotation "However, the knee joint does not have a physiological function. As more knowledge has become available, more and more PE teachers are abandoning the 'ring turn' in favour of the 'flexion and extension', which to some extent reduces the risk of injury due to irregularities and non-physiological functions. You are absolutely right in saying that the positive effects of health education on physical health are based on the willingness to learn and use them, but we are thinking of using new media to make health-related knowledge more widely available, thus increasing the number of "wobblers" who have received the knowledge but are still considering whether to learn and practice it. The number of "wavers" who have received the knowledge but are still considering whether to learn and practice it. It is a way of paving the way for a "quantitative change".

We have added an acknowledgement of WHO's work in the introduction and a discussion of the concerns you have raised in the discussion section.

Point 2: Various terms are used inconsistently throughout the paper – eg sport, medicine, public health and promoting health, it would be helpful for the reader to understand the definitions used by the authors for these important constructs. In addition a new term kinesiology is introduced in the conclusion, which further confuses. The term health promotion is used interchangeably with health education – they are different – review

Response 2: This part of your suggestion has been fully considered and we have decided to revise and describe the relevant words in the article more accurately. In addition, we would like to discuss with you the reasons for the presence of these terms in the article, which could easily lead to ambiguity among readers. In the Chinese academic world, the concept of "sports integration" has just been introduced, which refers to the combination of sports-related knowledge, such as the amount of activity, intensity of activity and technical specifications, with the medical aspects of the treatment of chronic diseases, in order to avoid the isolated and one-sided development of kinesiology and medicine.

Point 3Discussion on the population demographics of China, numbers of people, age groups, social group, disability, regional variations and access to the internet via for example smart phones/other devices would add weight to the arguments. Also which age groups/ societal groups are more likely to access sports/ health devices/apps. Has there been any evidence to show that people are having health benefit from using the app/ devices etc. Using the term the public is too misleading when discussing a population of 1.4 billion people. A clearer understanding of which group of people the study is reviewing is required. There is also an assumption that people are accessing health information but this is not explored or confirmed with evidence.

Response 3:

         We have described its characteristics and quantities. As of 2017, the number of Internet users in China reached 772 million, and this number is still growing. The number of Internet users in China is characterised by "more in the east and less in the west, more in the city and less in the countryside, and fewer young users than old ones"

         We have cited new references, but the data contained therein may be obscure to readers unfamiliar with transect surveys, and we can continue to add relevant data if you feel it is necessary.

         At the same time, you mentioned that it is not accurate to use the word "public" to cover 1.4 billion people, but in our humble knowledge, the word we can think of at the moment to express the concept of "mass" is public. Can you suggest a more accurate term?

Point 4: L. 54 is chemical reactions the most appropriate phrase to use here?

Response 4: We removed the inappropriate phrases

Point 5: L 60 – a critical discussion of SWOT would be useful early in the paper and a rationale for its choice.

Response 5: We thought it might be more in line with our thinking to introduce the choice of model for its analysis and why it was chosen, after the relevant social context had been described in the introduction. However. Given that you are more specialised than we are, where would you suggest we put the idea of a critical review of SWOT more specifically?

Point 6: L 80 – experts in what? Requires clarification.

Response 6: The issues here have been raised by other reviewers one after another, so we have explained them.

Point 7: L 258 – review use of word excellent – too emotive lacks supporting evidence

Response 7: The emotional adjective 'excellent' has been removed to maintain objectivity in the description.

Point 8: The work does not refer to the body of evidence on health literary

Response 8: Thank you for your question, we had not heard of the concept of 'health literature' before we did this research and it is not an area we specialise in.

Point 9What would regulation of the internet look like, how would it be achieved?

Response 9With regard to the ideal model of cyber regulation, we set out our vision in the discussion section, as follows. Thirdly, platforms strengthen the pre-qualification system for content creators and the post-qualification system for content regulation. Due to the special nature of sports and health-related content, unprofessional content can lead to poor user experience and even harm to the body. Therefore, the platform should recruit a certain number of profes-sional composite talents as content creators' qualification auditors, and set up a certain threshold audit system before the creators are admitted to the platform, so that only those who pass the audit can be admitted to the platform and obtain official certification. After the creator is on the platform, the auditors should regularly review the content published by the creator to review the professionalism of the content and try to avoid the dissemina-tion of wrong knowledge to avoid misleading the recipients of the content. However, at the same time, for non-professional creators who have already published other content and later want to transition to publishing health-related content, they should be given certain professional knowledge training and be audited according to their specific situation. A 'one-size-fits-all' type of auditing system will not only discourage people who want to ex-tend their content dissemination nodes , transforming the masses into communicators, and is not in line with the open and inclusive nature of new media.

Point 10: L 339 what percentage of the population using these apps/ devices engage in sport ? Clarity and objectivity required.

Response 10: In response to your suggestion, we have added specific reporting of changes in the relevant numbers at the end to increase objectivity. According to relevant reports, the scale of active users of sports and fitness apps rose rapidly to 89.28 million during the epidemic, nearly doubling year-on-year.

Point 11: L344 what were the searches around covid protection how did they relate to general fitness – more clarity required.

Response 11: As you say, the connection between the two is really not that deep, so we have rewritten and reworked this idea to be more logical. All things being equal, the susceptible population for COVID-19 epidemic is concentrated in the elderly and the sub-healthy, and this population is characterised by a weaker physical condition. According to research, people who are in better health have a lower rate of infection and a faster recovery rate from the disease. The example of the Newcastle pneumonia outbreak shows the role of good health in the early stages of a public health emergency as an outpost for disease prevention, and highlights the important role of good health and a healthy lifestyle in reducing the prevalence of more diseases. The need for health is not just a matter of increased awareness, but also a deepening of real needs. The "exercise card" model of apps such as Keep and the "online coaching" model of apps such as Tik Tok have created new forms of physical activity - relying on new media platforms. Home exercise on new media platforms not only alleviates the need for exercise and health demands of people at this particular time, but also provides a new type of exercise solution for the majority of people who do not have much free time to go to sports activities in the post-epidemic period.

Point 12: The stages of the swot analysis could be clearer, the reader would benefit from supported guidance at each step/ figure.

Response 12: We have recreated a flowchart (Figure 1), which can be found in the "Methods" section of the article.

Point 13: The conclusion assumes that ‘improvements’ would continue. What guidance is required based on the research?

Point 14: Where are the recommendations for the study?

Response 13 and 14:Public health encompasses many aspects; the increase in awareness of healthy living, the development of behavioural practices that promote physical health, the increase in sports participation and the orderly and controlled development of public health related indus-tries all reflect the building and improvement of public health in a region or country. The above perspectives involve macro control by the government and the platform and micro but extensive participation by the public. The three are interrelated as a whole and a prob-lem in any one link will hinder the overall development. According to the SWOT-AHP model, it can be seen that there are both strengths and threats to China's use of new media for public health. The huge internal strengths are not an excuse to ignore the threats, nor are the complex external threats simply a stumbling block to development; use the threats as a bottom line and maximise the strengths above the bottom line to expand the upper limit of development. Take advantage of the favourable policy and social environment, guard the red line of platform content regulation, and promote public health to ride on the new media fast track to sustainable development by taking public demands as a grip.

Supplementary Table 1.
Group
Sub-factors
Accept or Reject(√ OR ×)
Estimated intensity
Strengths
(S)
S1

1-5 (The higher the absolute value of the score, the more important it is)
S2

S3

S4

Opportunities(O)
O1

O2

O3

O4

Weakness
(W)
W1

(-1)-(-5) (The higher the absolute value of the score, the more important it is)
W2

W3

Threats
(T)
T1

T2

T3

Supplementary Table 2.
Using S group to explain.
Comparisons among subfactors within group S

Equal Importance
Moderate importance
Strong importance
Very strong or demonstrated importance
Extreme importance
Importance between the above levels
If sub-factor A is less important than sub-factor B, then Intensity of Importance of A is the inverse of Intensity of Importance of B

1
3
5
7
9
2,4,6,8
1/2,1/3,1/4,1/5,1/6,1/7,1/8,1/9

S1

S1
S1

S2
S1

S3
S1

S4
S2

S1
S2

S2
S2

S2
S2

S2
S3

S3
S3

S3
S3

S3
S3

S3
S4

S4
S4

S4
S4

S4
S4

S4

Round 2

Reviewer 3 Report

2nd review 

Thank you for your amendments.  I was surprised that the amendments were made so quickly. Unfortunately, many of the amendments do not demonstrate an understanding of the concepts used in the paper. This work attempts to cover a very broad area but in doing so it covers the concepts and theories very superficially. At this international level readers expect to see an in-depth understanding of the different areas covered by the paper. The work would be improved if it could demonstrate a more thoughtful appreciation of the facets of behaviour change, which go beyond merely the accumulation of knowledge or information. As highlighted previously it also needs to define the key topics it seeks to discuss – ie sport and medicine. Are you referring to sports medicine, which is a branch of medicine which promotes fitness and the prevention of injuries related to sport or a more global definition of the terms? Or is this a term used only in China? Its really important to provide accuracy and objectivity when writing for peers at this level.

In addition it needs to be made clear at the beginning of this paper that the work centres on China and its population. Is it possible for the paper to make global statements when many of the areas covered relate only to China?

L34 – review word choice – sub-healthy (this appears to be a term used in China) not appropriate, if it is to be used requires a definition – too vague. Perhaps the paper is  referring to sub-optimal health status?

L35 – where is the evidence for this statement.

L36 – lacks clarity, is not objective and grounded in evidence. Does not demonstrate an understanding of these concepts and their impact on health behaviour. Review.

L39 in conjunction with other important factors – suggest reread cited paper

L40 what is sport and medicine? Not clear

L65 – without a critique it is not possible to state that it  – ‘provides a good role’ .

L76 the paper provides a description of swot but no critique – it does not offer a justification for its use in this research – there is a wealth of material in the field of business offering a critique of swot.

 L382 – review word choice – sub-healthy not appropriate

383 – what research?

L380 to L395 very descriptive writing – lacks critique as required at this level

L584 requires rewording

L86 there is a lack of understanding of the concept of health literacy – review. I am including, merely as an example, two papers on health literacy – there is an extensive body of knowledge and research on this topic. 

 Health literacy, is defined as the ability to access, understand, appraise and apply health information. Merely having health information does not guarantee behaviour change - this important concept is not covered in this paper. 

L594 – such as?

L600 kinesiology still remains in the conclusion and has not been raised or discussed elsewhere in the work. This comments was raised previously.

Author Response

Response to Reviewer 1 Comments

Point 1 and Point 8: L34 – review word choice – sub-healthy (this appears to be a term used in China) not appropriate, if it is to be used requires a definition – too vague. Perhaps the paper is  referring to sub-optimal health status? AND L382 – review word choice – sub-healthy not appropriate

Response 1: In the mid-1980s, Professor Buchman of the Soviet Union discovered that in addition to the healthy and diseased states, there is an intermediate state of the human body that is not healthy or diseased, called the subhealth state. A global survey by the WHO showed that only 5% of people are truly healthy, 20% are sick, and 75% are in a state of subhealth.

         We have added some definitions of "subhealth" to the article, which is a combination of the Web Encyclopedia and highly cited papers.

         The third state, also known as the grey state, pre-morbid state, sub-clinical stage, pre-clinical stage and latent disease stage, includes the absence of clinical symptoms or the perception of minimal symptoms, but with underlying pathological information. This concept is more popular in Chinese academia and society, and is considered by scholars to be similar to the "chronic fatigue syndrome" developed by the US Centers for Disease Control and Prevention.

         However, due to your professionalism, we would be very happy to follow your advice and replace "sub-health" with your proposed "sub-optimal health status" if you feel that we have not explained it clearly.

Point 2 and Point3 : L35 – where is the evidence for this statement. AND L36 – lacks clarity, is not objective and grounded in evidence. Does not demonstrate an understanding of these concepts and their impact on health behaviour. Review.

Response 2 and 3: I am sorry that this idea, which we have rightly assumed to be a consensus and written in the paper, has been introduced into the new literature at your suggestion to provide relevant evidence.

Point 4: L39 in conjunction with other important factors – suggest reread cited paper:

Response 4: This sentence does lack evidence to a certain extent, so we decided to remove it.

Point 5: L40 what is sport and medicine? Not clear

Response 5: a branch of medicine which promotes fitness and the prevention of injuries related to sport. As you said in the previous article, "the combination of sport and medicine" or "sport and medicine" as it may appear in the text means "a branch of medicine which promotes fitness and the prevention of injuries related to sport", the concept we describe is more popular in China, but due to the language barrier, it may not be possible to express what we want to say accurately in English, please understand.

Point 6: L65 – without a critique it is not possible to state that it  – ‘provides a good role’ .

Response 6: Yes, as you say, we have removed the modifier "good" from the role.

Point 7: L76 the paper provides a description of swot but no critique – it does not offer a justification for its use in this research – there is a wealth of material in the field of business offering a critique of swot.

Response 7: Yes, as you say, without discussing the shortcomings of the SWOT model, the logic of the text would be somewhat flawed (it would be impossible to account for why we chose the SWOT-AHP model instead of the SWOT model), so we have added a critical perspective on it.

         Although the analysis successfully pinpoints the factors, individual factors are usually described briefly and very generally. For this reason, SWOT analysis possesses deficiencies in the measurement and evaluation steps. Although the analytic hierarchy process (AHP) technique removes these deficiencies, it does not allow for measurement of the possible dependencies among the factors.

Point 9: 383 – what research?

Response 9: We have added citations.

Point 10: L584 requires rewording

Response 10: What we want to talk about here is the "gripper".

Point 11: L86 there is a lack of understanding of the concept of health literacy – review. I am including, merely as an example, two papers on health literacy – there is an extensive body of knowledge and research on this topic. 

Response 11: Thank you very, very much for your advice, I think that's what peer review is for, and we will continue to drill down into related areas to add to our deficiencies after the work on this article is over, and hopefully we can follow your example and be able to follow in your footsteps faster in this area. It is true that health literacy is a crucial issue, but this article is more about the conditions that prepare the public for health literacy: the use of new media to disseminate relevant content. The issue of health literacy does require greater and longer-term attention at a societal level.

Point 12: L594 – such as?

Response 12: For example, the author ignores the issue of "health literacy" in the construction of the SWOT model (Health literacy, is defined as the ability to access, understand, appraise and apply health information. Merely having health information does not guarantee behaviour change)

Point 13: L600 kinesiology still remains in the conclusion and has not been raised or discussed elsewhere in the work. This comments was raised previously.

Response 13: We are talking here about the combination of media, sport and medicine.

Point 14: Thank you for your amendments.  I was surprised that the amendments were made so quickly. Unfortunately, many of the amendments do not demonstrate an understanding of the concepts used in the paper. This work attempts to cover a very broad area but in doing so it covers the concepts and theories very superficially. At this international level readers expect to see an in-depth understanding of the different areas covered by the paper. The work would be improved if it could demonstrate a more thoughtful appreciation of the facets of behaviour change, which go beyond merely the accumulation of knowledge or information. As highlighted previously it also needs to define the key topics it seeks to discuss – ie sport and medicine. Are you referring to sports medicine, which is a branch of medicine which promotes fitness and the prevention of injuries related to sport or a more global definition of the terms? Or is this a term used only in China? Its really important to provide accuracy and objectivity when writing for peers at this level.

In addition it needs to be made clear at the beginning of this paper that the work centres on China and its population. Is it possible for the paper to make global statements when many of the areas covered relate only to China?

reply 14:

Your comments are very pertinent, but it may be that we are limited by our circle of people to communicate more deeply with experts around the world that prevents a deeper understanding.

  The other two comments you have made have been annotated or supplemented in the text in the hope that they will help you to read it more smoothly.
